# Grape-RNA: A Database for the Collection, Evaluation, Treatment, and Data Sharing of Grape RNA-Seq Datasets

**DOI:** 10.3390/genes11030315

**Published:** 2020-03-16

**Authors:** Yi Wang, Rui Zhang, Zhenchang Liang, Shaohua Li

**Affiliations:** 1Beijing Key Laboratory of Grape Science and Enology, and CAS Key Laboratory of Plant Resources, Institute of Botany, the Innovative Academy of Seed Design, the Chinese Academy of Science, Beijing 100093, China; wangyi19881107@163.com (Y.W.); shhli@ibcas.ac.cn (S.L.); 2University of Chinese Academy of Sciences, Beijing 100049, China; 3College of Plant Protection, Shandong Agricultural University, Taian 271018, China; ruizhangly@163.com; 4Sino-Africa Joint Research Center, Chinese Academy of Sciences, Wuhan 430074, China

**Keywords:** Grape-RNA, RNA-seq, database, web tools

## Abstract

Since its inception, RNA sequencing (RNA-seq) has become the most effective way to study gene expression. After more than a decade of development, numerous RNA-seq datasets have been created, and the full utilization of these datasets has emerged as a major issue. In this study, we built a comprehensive database named Grape-RNA, which is focused on the collection, evaluation, treatment, and data sharing of grape RNA-seq datasets. This database contains 1529 RNA-seq samples, 112 microRNA samples from the public platform, and 485 RNA-seq in-house datasets sequenced by our lab. We classified these data into 25 conditions and provide the sample information, cleaned raw data, expression level, assembled unigenes, useful tools, and other relevant information to the users. Thus, this study provides data and tools that should be beneficial for researchers by allowing them to easily use the RNA-seq. The provided information can greatly contribute to grape breeding and genomic and biological research. This study may improve the usage of RNA-seq.

## 1. Introduction

Gene expression is one of the most important processes in life activities, playing a vital role in the development and environmental adaptation of living things. The study of gene expression is fundamental for functional genomics research, especially in the post-genomic era, which began with the completion of the Human Genome Project in 2001 [1]. Unlike previous technologies, RNA sequencing (RNA-seq) can survey the spatial and temporal expression level of a whole genome at one time based on high-throughput sequencing technologies [2]. The expression levels of all expressed genes can be calculated accurately according to the number of reads. Therefore, RNA-seq has become the most popular method used by studies on expression levels in the life sciences.

RNA-seq has been used in almost all areas of plant science, such as in plant development, gene regulation analysis, abiotic and biotic resistance, plant breeding and cultivation, and plant evolution and diversity analysis [3,4,5,6]. For example, on the basis of the RNA-seq of some special development and ripening stages, some putative key genes that can regulate plant development were identified [4,7,8]. The RNA-seq data of overexpression plants can present new insights into the regulation network of many genes [6,9,10]. A comparison of the RNA-seq datasets of plants under different abiotic or biotic stresses can yield candidate genes related to these stresses [5,11,12,13]. Breeders can explore the inherent rules of plant breeding and cultivation according to the expression status of the RNA-seq in different plants or areas [14]. In addition, the domestication of cultivars, evolution relationships, and genome duplications of the plant genome can be identified by the assembled genes and phylogenetic analysis with transcriptome datasets [14,15,16,17]. Thus, given its numerous usages and advantages, RNA-seq has become a routine tool for plant sciences, and the datasets of RNA-seq have been continuously increasing in the last few years. However, most studies on RNA-seq have mainly focused on a few special phenotypes or target genes, and abundant useful information is missing. The full utilization of these RNA-seq datasets has become an urgent requirement for researchers. The most effective way to solve this problem is to establish user-friendly databases that can collect, treat, and share relevant data and results comprehensively. Many high-efficiency software and web-based platforms on RNA-seq and microarray have been developed and established in the last few years [18,19,20,21].

Grape is a widely cultivated fruit crop worldwide and has great economic value. Similar to other plants, numerous transcriptome studies on grape have been conducted, which have produced a number of RNA-seq datasets [4,22]. The first RNA-seq of grape was published in 2010 on the berry development of *Vitis vinifera* [23]. Since then, a series of RNA-seq experiments have been performed. Palumbo et al. (2014) published the RNA-seq data of 60 samples in one article to elucidate the ripening of grape berry [4]. The correlation between gene expression and berry skin anthocyanin accumulation was established in 2017 by using data from 120 samples of the RNA-seq of red and white grapevine varieties [22]. In addition to these experiments, numerous studies have provided a large number of RNA-seq datasets [5,24]. Until 2018, approximately 1500 high-quality RNA-seq datasets have been deposited in public repositories.

To enhance the utilization of these RNA-seq datasets, Moretto et al. established a database named Vespucci, which provides the normalized expression levels of 1608 samples of microarray and RNA-seq datasets [19]. This database greatly contributed to improving grape sciences. However, Vespucci is focused on the microarray data and includes only 135 RNA-seq datasets. In addition, this database only provides the normalized expression data by COLOMBOS, and it does not provide additional underlying data, such as the cleaned raw data, unigenes, and especially the fragments per kilobase per million (FPKM) value of each gene. In addition, in the past three years, many new RNA-seq datasets have been published. A new professional database on grape RNA-seq should be established. This database should be richer, more accurate, comprehensive, and more user-friendly.

In this study, we collected all 1641 published RNA-seq datasets and 485 RNA-seq datasets from our lab. The published datasets included 1529 RNA-seq samples and 112 microRNA samples from the public repositories deposited before 2018. The collected data contained more than 19 *Vitis* species and approximately 130 varieties under 25 conditions. All public and in-house data were treated according to a standard pipeline, and all results were deposited into the Grape-RNA database (http://www.grapeworld.cn/gt/index.html). Grape-RNA provides high-quality raw data, expression level, unigenes of each variety, co-expression network, other useful tools (e.g., BLAST+, BLAT, and SignalP), and other relevant information services.

## 2. Materials and Methods

### 2.1. Data Source

All public data were downloaded from the National Center for Biotechnology Information (NCBI) Sequence Read Archive (SRA) database. Detailed information is provided in Appendix A (please see Appendix A). The in-house data were sequenced by our lab and have not been published. Detailed information can also be found in the database.

### 2.2. Data Treatment 

The RNA-seq and microRNA data in SRA format were transformed into FASTQ format by FASTQ-DUMP (2.6.3, https://github.com/ncbi/sra-tools/wiki), and all the RNA-seq data were quality controlled by Trimmomatic (v0.36) [25] with the parameter of LEADING:3 TRAILING:3 SLIDINGWINDOW:4:15 MINLEN:30. The data were aligned to the 12X genome [26] by HISAT2 (v2.1.0) with the default parameters [27], and the expression levels were obtained by StringTie (v1.3.3b) [27] based on the annotation version of Genoscope12X [26] and IGGP V2.1 [28] (Figure 1). Samples with mapping rates lower than 60% were marked as problematic data and not considered in the database. The microRNA data were processed by miRDeep2 (v2.0.0.8) [29] through adaptor clipping, expression level calculation, and secondary structure identification. The minimum length of the clipped sequence were 18 nt and the used microRNA database of 186 matured microRNAs were downloaded from miRbase (http://www.mirbase.org/).

### 2.3. Development of Data Mining Tools 

Data mining tools were developed based on a number of different software programs. BLAST+ (v2.6.0+) [30] and BLAT (v.35) [31] were used for sequence alignment. Gene Ontology (GO) annotation was obtained by eggNOG (v1.0.3) [32] using the database of eukaryotes. KEGG annotation was processed by the KEGG Automatic Annotation Server (KAAS, www.genome.jp/tools/kaas/) web tools in the KEGG database (www.kegg.jp) [33]. The signal peptides of the grape were identified by SignalP (v4.1) [34] using the default parameters. Co-expression analysis was processed by WGCNA (v1.63) [35], The threshold in the co-expression analysis was 0.9. Co-expression analysis provides users with the network information, and the exported co-expression network data can be displayed by Cytoscape [36]. In this database, data visualization of expression level or other information was mainly realized by R. The heatmap tool in this database was also processed by R, and providing some versatile parameters, the result was provided in PNG and PDF. Besides this, other tools also provided some versatile parameters for users.

### 2.4. Database Architecture and Web Interface

All obtained data, including the cleaned raw data of RNA-seq and microRNA-seq, as well as the processed results of this database, were stored in Grape-RNA. This database was constructed on a Centos6.5 platform by using HTML5, Mysql, PHP, and Perl scripts. The web interface was written in HTML5 and CSS3. The webtools were based on several Perl scripts and some open source software.

## 3. Results and Discussion

### 3.1. Data Source and Statistics

A total of 2126 RNA-seq grape datasets were used in this database, of which 1641 were public data in SRA format downloaded from the NCBI (https://www.ncbi.nlm.nih.gov/), and 485 were unpublished in-house RNA-seq samples. The public data consisted of 1529 RNA-seq samples and 112 microRNA samples. The database also included 43.61 G RNA-seq reads, and the average size of the data was 28.52 M reads/fragments per sample (Figure 2A). These data belong to 85 experiments, 16 tissues, and 25 different treatment conditions (Appendix A). Sample information can be found in the sample page (http://www.grapeworld.cn/gt/sample.html). Overall, the data of the Grape-RNA database cover the majority of fields in grape science.

### 3.2. RNA-Seq and MicroRNA Data Processing

To obtain the accurate expression levels and make them suitable for different studies, all data were processed by a standard pipeline (Figure 1). Raw RNA data were cleaned by Trimmomatic (v0.36) [25], and then the filtered clean data were mapped on the grape genome sequence (12X, Genoscope) [26] using HISAT2 (v2.1.0) [27]. Samples with mapping rates lower than 60% were marked as problematic data and not considered in the database. On average, 88.52% of reads were mapped on the reference genome (Figure 2B), and the map rate of 91.59% of the samples was higher than 80%. Afterwards, two classical and widely used annotation versions, namely, Genoscope12X [26] and IGGPV2.1 [28], were used, and the expression level (FPKM) was obtained by StringTie (v1.3.3b) [27]. When only the genes with FPKM ≥ 1 were considered, only an average of 9482 genes could be identified (Figure 2C). When this threshold was decreased to 0.05, 12,573 genes could be identified (Figure 2D). All these expression levels and other relevant information were stored in the database (http://www.grapeworld.cn/gt/exp.html). 

The microRNA data were processed by miRDeep2 (v2.0.0.8) [29], and 186 matured microRNAs from miRBase (http://www.mirbase.org/) were used as reference. The de novo miRNAs of each sample were also deposited in this database. All microRNA data, including expression level, mature sequence, precursor sequence, and other relevant information, were deposited in the database (http://www.grapeworld.cn/gt/exp.html).

### 3.3. Database Architecture and Data Mining Tools

To increase the accessibility of the expression and other information, Grape-RNA (http://www.grapeworld.cn/gt/index.html) was developed by using searching, versatile data mining, and data visualization tools based on LAMP, Perl, and R via a user-friendly interface (Figure 3).

#### 3.3.1. Searching Tools

Several searching tools were developed in this database (http://www.grapeworld.cn/gt/transcriptome.html), such as sample search for RNA-Seq and microRNA-seq, raw data search and download, and expression level search. In regard to the sample searching tools, users can search for the sample information according to the tissue and treatment of the experiment. The result page provides detailed information, such as sample ID, NCBI accession, experiment description, and varieties. On this page, people can search the targeted samples by tissue or treatment, people also can search the sample of treatment on special tissues. For example, there are 60 samples related to heat stress, 1096 samples related to berry or berry skin, and only 48 of the samples are related to heat stress on the grape berry. The search page for the raw data requires only the sample IDs and provides the download link of all the corresponding raw data files. Two essential forms of data, gene ID [12X (GSVIVT) or IGGPv2.1 (VIT)] and sample ID (GWGT), are necessary to search for the expression level of RNA-seq. The expression level of the required genes will be listed in table format on the result page. For example, if we want to know the expression level of several genes during heat stress on grape berry, the gene ID and sample ID of 48 samples related to heat stress on grape berry should be input, and the expression level of these genes will be searched and provided on the results page. Users can also download the expression information file from a download link. This expression file can be used as the input files for the heatmap tools. Expression level of microRNA only requires the sample ID, and all microRNA information of this sample will be listed in the result page. In addition to these three search tools, we provide the search tools of GO, KEGG, NR annotation, and some related papers. These tools can supply multiple types of functional annotation of the targeted genes and give some guidance to the users, and also can do benefit to other works, such as RNA-seq analysis and gene identification.

#### 3.3.2. Data Mining Tools

Grape-RNA includes abundant tools to make the research of related studies more convenient (http://www.grapeworld.cn/gt/tools.html). Some of these tools are BLAST, BLAT, GO and KEGG annotation, signal peptide identification, transmembrane finder, RT-qPCR primer design, sequence tools, and microRNA regulatory network. These tools were developed based on all existing annotation versions of 8X and 12X grape genome, and supplied the information of similar sequences, high quality primers, or the information of signal peptide, transmembrane domain, and microRNA regulation. These tools can help researchers avoid the need for complex programming languages and a huge number of computations, making the undertaking of related studies easier and more efficient. In addition, 1529 high-quality samples were used in the co-expression analysis. This analysis was based on the IGGPv2.1 annotation version and processed by WGCNA (v1.63) [35]. The threshold in the co-expression analysis was correlation coefficient > 0.9, and 15,291 high-quality genes were considered. Finally, 35 edges and nodes were identified according to the expression level. In this tool, the user can input the gene ID and obtain the potential related genes. Co-expression analysis will also provide users with the network information, and the exported co-expression network data are displayed by Cytoscape [36]. For example, if we want to know the co-expression relationship of VIT_214s0081g00540.1, we just input this gene ID into the text area, and finally we find that 177 candidate genes were involved in this network. Data visualization in this database was mainly realized by the use of R. The heatmap tools provide users with versatile parameters, facilitating the generation of the desired heatmap figures, and the result will be provided in PNG and PDF.

### 3.4. Procedure for Using Grape-RNA

In Grape-RNA, we provide a help page (http://www.grapeworld.cn/gt/help.html) to allow users to quickly begin using this database. We provide detailed instructions with a screenshot for each section. All introduction files are in PDF and can be accessed freely through the network.

### 3.5. Perspectives

Grape-RNA is the most comprehensive RNA-seq database for grape studies. This database considers all annotation versions and provides the expression of the two most used versions. The samples in the database are representative and cover the majority of areas in recent studies. Grape-RNA also provides a number of utilities to enhance the accessibility, clarity, and accuracy of the data. This database will improve the efficiency of utilizing the RNA-seq data and greatly contribute to grape studies.

Grape-RNA is a part of Grapeworld (www.grapeworld.cn) and can be accessed worldwide at any time. The database will be updated when enough new sequenced RNA-seq data have been collected or a new analysis method emerges.

Grape-RNA is currently the most comprehensive and versatile grape RNA-seq database and provides a user-friendly interface. Compared with VitisNet [37] and VESPUCCI [19], Grape-RNA contains more RNA-seq datasets, and classifies the samples properly. Grape-RNA also provides accurate whole genome expression levels of more than 2,000 samples and provides some related tools and information. We believe that this database can greatly contribute to grape breeding and genomic and biological research. This study may improve the usage of RNA-seq.

## Figures and Tables

**Figure 1 genes-11-00315-f001:**
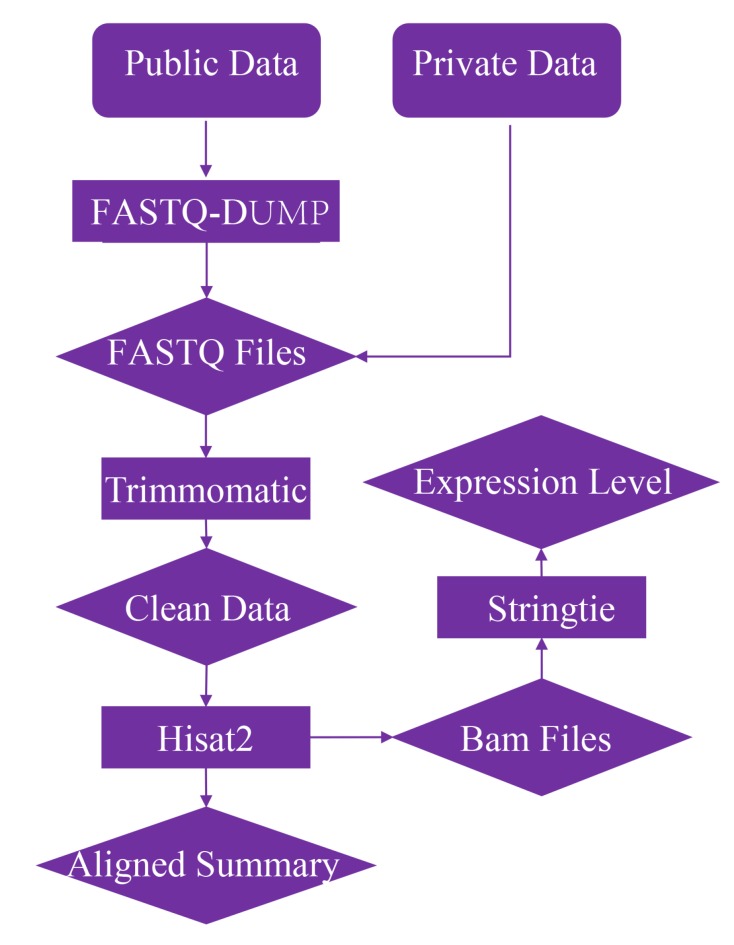
The standard pipeline of the treatment of public and private RNA-seq data.

**Figure 2 genes-11-00315-f002:**
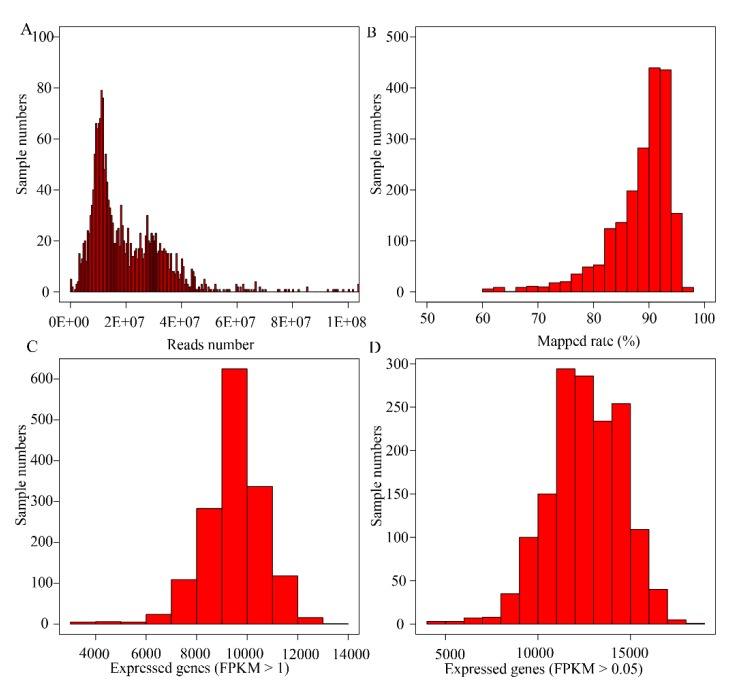
A landscape of the mapping data and expression. (**A**) Reads number distribution of all samples in Grape-RNA. (**B**) Mapping rate distribution of all samples in this database. (**C**) Gene number (FPKM > 1) distribution of all these samples. (**D**) Gene number (FPKM > 0.05) distribution of all these samples.

**Figure 3 genes-11-00315-f003:**
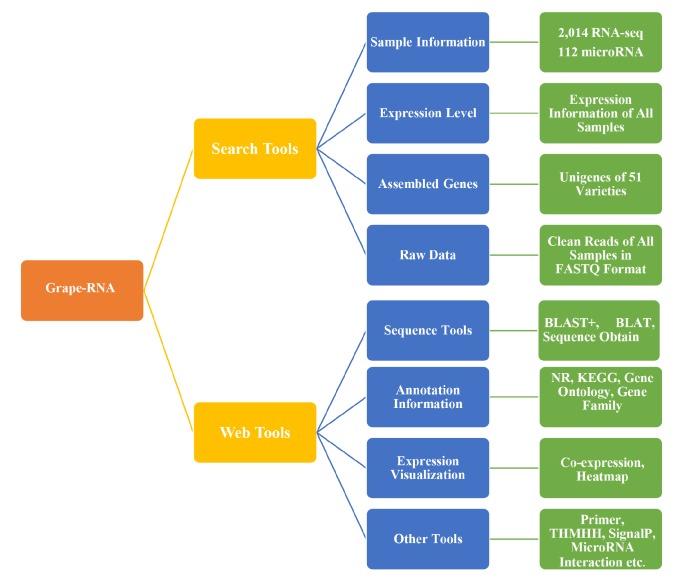
The detail information and architecture structure of Grape-RNA.

## Data Availability

The datasets that were generated and/or analyzed in this study are available in Grape-RNA (http://www.grapeworld.org/gt/index.html), and the raw data were stored in the database (http://www.grapeworld.cn/gt/rawdata.html). The 485 in-house RNA-seq datasets have been deposited in NCBI Sequence Read Archive (SRA) under BioProject accession PRJNA565689 and PRJNA554706.

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
