# Peer review of "Grape-RNA: A Database for the Collection, Evaluation, Treatment, and Data Sharing of Grape RNA-Seq Datasets"

_genes, 2020, doi:10.3390/genes11030315_

Round 1

Reviewer 1 Report

The manuscript describes the novel database covering grape transcriptome. The collected data and pipelines are clearly presented, and what's most important - the database works, though it is hard to navigate and requires prior knowledge of sample set structure.

The main concern is the readability of the text. I'm not a native speaker and my expertise in the English language is rather limited, but I do believe that this manuscript could use some additional proof-reading.

The second important issue regards data availability. The transcriptomes of 485 samples were obtained by authors. Though the raw data are deposited in the database, according to journal guidelines, "New high throughput sequencing (HTS) datasets (RNA-seq, ChIP-Seq, degradome analysis, …) must be deposited either in the GEO database or in the NCBI’s Sequence Read Archive". Thus, authors should provide a SRA or GEO accessions for raw data.

Minor comments:

Line 59: Vitis vinifera - species nomenclature should be in italic.

Line 89 and below: "Stable 1" is an unusual abbreviation for a Supplementary table. According to the journal guidelines, supplementary tables should be indicated as "Table S1".

Author Response

Comments and Suggestions for Authors

The manuscript describes the novel database covering grape transcriptome. The collected data and pipelines are clearly presented, and what's most important - the database works, though it is hard to navigate and requires prior knowledge of sample set structure.

Question 1: The main concern is the readability of the text. I'm not a native speaker and my expertise in the English language is rather limited, but I do believe that this manuscript could use some additional proof-reading.

Reply 1: We have modified the typos/grammar errors and ask for a professional language services to fix the language problem.

Question 2: The second important issue regards data availability. The transcriptomes of 485 samples were obtained by authors. Though the raw data are deposited in the database, according to journal guidelines, "New high throughput sequencing (HTS) datasets (RNA-seq, ChIP-Seq, degradome analysis, …) must be deposited either in the GEO database or in the NCBI’s Sequence Read Archive". Thus, authors should provide a SRA or GEO accessions for raw data.

Reply 2: We now have added the accession numbers of PRJ in the “Availability of data and materials” section.

Question 3: Line 59: Vitis vinifera - species nomenclature should be in italic.

Reply 3: Thanks very much for your carefully work, we have modified this mistake and checked all species nomenclatures.

Question 4: Line 89 and below: "Stable 1" is an unusual abbreviation for a Supplementary table. According to the journal guidelines, supplementary tables should be indicated as "Table S1".

Reply 4: Sorry for this mistake, we have checked the whole manuscript and modified all mistakes.

Reviewer 2 Report

The manuscript titled “Grape-RNA: a database for the collection, evaluation, treatment and data sharing of grape RNA-seq datasets” reports the description of a comprehensive RNA-seq database for Vitis. I really appreciate efforts the authors made to collect data of more than 2000 samples and to include them in the Grape-RNA database.

Here’s a list of parts that need to be revised.

- Page 2, line 72. “In the past three years, many new RNA-seq datasets have been published. A new professional database on grape RNA-seq should be established. This database should be richer, more accurate, comprehensive, and more user-friendly compared with current databases.” Which one they are? Please, clarify why a new database should be designed. What do you mean with “accurate”?

- The paragraph “Data treatment” and “Development of data mining tools” need to be described better. Please, include information about the parameters used. Moreover, in Results and Discussion section there are several information that need to be included in Material and Methods section: i) “Samples with mapping rates lower than 60% were marked as problematic data and not considered in the database.”; ii) “two widely used annotation versions, namely, Genoscope12X and IGGPV2.1 [34], were used”; iii) “and 186 matured microRNAs from the miRBase (http://www.mirbase.org/) were used as the reference”; iv) “This analysis was based on the IGGPv2.1 annotation version and processed by WGCNA (v1.63)[35]. The threshold in the coexpression analysis was 0.9”; v) Data visualization in this database was mainly realized by R. The heatmap tools provide users with versatile parameters, facilitating the generation of the desired heatmap figures, and the result will be provided in PNG and PDF. Co-expression analysis will also provide users with the network information, and the exported co-expression network data are displayed by Cytoscape”.

- Page 4, line 129. “two widely used annotation versions, namely, Genoscope12X and IGGPV2.1 [34], were used”. Why?

- Page 5, line 154. “Two essential data, gene ID [12X (GSVIVT) or IGGPv2.1 (VIT)] and sample ID (GWGT), are necessary to search for the expression level”. Please, specify that both gene ID and sample ID are necessary only for RNA-seq data and not for microRNA data.

- Page 5, line 148. “In addition to these three search tools, we provide the 158 search tools of GO, KEGG, NR annotation, and some related papers”. Please, provide information about the output these tools produce and why they are useful of the user.

- Page 5, line 162. “Some of these tools are BLAST, BLAT, GO and KEGG annotation, signal peptide identification, transmembrane finder, RT-qPCR primer design, sequence tools, and microRNA regulatory network”. Please, provide information about the output these tools produce and why they are useful of the user.

- Please, include some cases of study for the expression level tools and for the coexpression analysis, highlighting the usefulness of this database.

- The discussion is almost non-existent. Please, discuss how Grape-RNA differs from the other database, such as VESPUCCI or VitisNet.

Minor revision:

- Page 2, line 72. “FPKM”: abbreviation to be defined.

- Page 2, line 83. “Grape-RNA is currently the most comprehensive and versatile grape RNA-seq database and provides a user-friendly interface. We believe that this database can greatly contribute to grape breeding and genomic and biological research. This study may improve the usage of RNA-seq”. Please, remove from the introduction, this is a conclusion.

- Page 2, line 89. “Stable 1”. Please, replace with Table S1.

- Page 3, line 94. “12X genome”. Please, include reference.

- I suggest to rearrange the figure enumeration. Please, move the Figure 2 (The standard pipeline of the treatment of public and private RNA-seq data.) from results to material and methods section “Data treatment”.

- Page 4, line 119. “Stable 1”. Please, replace with Table S1.

- Page 4, line 132. “All these expression levels and other relevant information were stored in the database”. Where? how I access these information? Please, could you provide a screenshot as supplementary material?

- Page 4, line 138. “All microRNA data, including expression level, mature sequence, precursor sequence, and other relevant information, were deposited in the database”. Where? how I access these information? Please, could you provide a screenshot as supplementary material?

- Please, revise carefully the references. Reference 25, 28 refer to supplementary materials of Reference 24 and 27.

Author Response

Comments and Suggestions for Authors

The manuscript titled “Grape-RNA: a database for the collection, evaluation, treatment and data sharing of grape RNA-seq datasets” reports the description of a comprehensive RNA-seq database for Vitis. I really appreciate efforts the authors made to collect data of more than 2000 samples and to include them in the Grape-RNA database.

Here’s a list of parts that need to be revised.

Question 1: Page 2, line 72. “In the past three years, many new RNA-seq datasets have been published. A new professional database on grape RNA-seq should be established. This database should be richer, more accurate, comprehensive, and more user-friendly compared with current databases.” Which one they are? Please, clarify why a new database should be designed. What do you mean with “accurate”?

Reply 1: Sorry for the misleading, here we want to mention that the increase of RNA-seq need a new database, this database should provide all useful and accurate information of all grape RNA-seq datasets. Here, “accurate” means abandon the low quality datasets, and treated the RNA-seq datasets in a standard pipeline.

Question 2: The paragraph “Data treatment” and “Development of data mining tools” need to be described better. Please, include information about the parameters used. Moreover, in Results and Discussion section there are several information that need to be included in Material and Methods section: i) “Samples with mapping rates lower than 60% were marked as problematic data and not considered in the database.”; ii) “two widely used annotation versions, namely, Genoscope12X and IGGPV2.1 [34], were used”; iii) “and 186 matured microRNAs from the miRBase (http://www.mirbase.org/) were used as the reference”; iv) “This analysis was based on the IGGPv2.1 annotation version and processed by WGCNA (v1.63)[35]. The threshold in the coexpression analysis was 0.9”; v) Data visualization in this database was mainly realized by R. The heatmap tools provide users with versatile parameters, facilitating the generation of the desired heatmap figures, and the result will be provided in PNG and PDF. Co-expression analysis will also provide users with the network information, and the exported co-expression network data are displayed by Cytoscape”.

Reply 2: Thanks for this suggestion, we have added these information in the right section of this article and we also added some other useful information.

Question 3: Page 4, line 129. “two widely used annotation versions, namely, Genoscope12X and IGGPV2.1 [34], were used”. Why?

Reply 3: This two annotation version was classical and wildly used in grape studies, and can cover most of the grape studies. And other annotation version can find the corresponding IDs of these two version in this database. We have revised the expression here to make it more acceptable.

Question 4: Page 5, line 154. “Two essential data, gene ID [12X (GSVIVT) or IGGPv2.1 (VIT)] and sample ID (GWGT), are necessary to search for the expression level”. Please, specify that both gene ID and sample ID are necessary only for RNA-seq data and not for microRNA data.

Reply 4: We have revised this section and specified the gene ID and sample ID only necessary for the expression level of RNA-seq in line 170. we also added a sentence to illustrate method of microRNA data in line 175-176.

Question 5: Page 5, line 148. “In addition to these three search tools, we provide the search tools of GO, KEGG, NR annotation, and some related papers”. Please, provide information about the output these tools produce and why they are useful of the user.

Reply 5: Thanks for the suggestion, we have added these information in line 177-179. And detail information also can be found in the help page of this database.

Question 6: Page 5, line 162. “Some of these tools are BLAST, BLAT, GO and KEGG annotation, signal peptide identification, transmembrane finder, RT-qPCR primer design, sequence tools, and microRNA regulatory network”. Please, provide information about the output these tools produce and why they are useful of the user.

Reply 6: We have discussed the output and useful of these tools in line 185-188. And detail information how to use these tools and the output also can be found in the help page of this database.

Question 7: Please, include some cases of study for the expression level tools and for the coexpression analysis, highlighting the usefulness of this database.

Reply 7: Thanks for this suggestion, we have added a small case in this article to make the sample search(line164-167), expression (line 170-173) and co-expression analysis (line 193-197) more useful.

Question 8: The discussion is almost non-existent. Please, discuss how Grape-RNA differs from the other database, such as VESPUCCI or VitisNet.

Reply 8: we have discussed this in the last of the article (line 214-219), and addressed the difference and important of this database.

Question 9: Page 2, line 72. “FPKM”: abbreviation to be defined.

Reply 9: Thanks for you carefully work, we have added the full name of FPKM (Fragments Per Kilobase per Million) in this place.

Question10: Page 2, line 83. “Grape-RNA is currently the most comprehensive and versatile grape RNA-seq database and provides a user-friendly interface. We believe that this database can greatly contribute to grape breeding and genomic and biological research. This study may improve the usage of RNA-seq”. Please, remove from the introduction, this is a conclusion.

Reply 10: Yes, this section is not proper here, we have removed it and re-write it in the last of the article (line 214-219).

Question 11: Page 2, line 89. “Stable 1”. Please, replace with Table S1.

Reply 11: Sorry for this mistake, we have checked the whole article and modified this mistakes.

Question 12: Page 3, line 94. “12X genome”. Please, include reference.

Reply 12: We have added this reference in the right place and rearranged the reference.

Question 13: I suggest to rearrange the figure enumeration. Please, move the Figure 2 (The standard pipeline of the treatment of public and private RNA-seq data.) from results to material and methods section “Data treatment”.

Reply 13: We have moved it.

Question 14: Page 4, line 119. “Stable 1”. Please, replace with Table S1.

Reply 14: We have checked the whole manuscript and modified this mistakes.

Question 15: Page 4, line 132. “All these expression levels and other relevant information were stored in the database”. Where? how I access these information? Please, could you provide a screenshot as supplementary material?

Reply 15: Thankss very much for pointing out this problem, we have added the URL to the article (line 146), and as the reviewers advised, the screenshot is very important, so we provided all screenshot for all tools in the help page in this database (http://www.grapeworld.cn/gt/help.html).

Question 16: Page 4, line 138. “All microRNA data, including expression level, mature sequence, precursor sequence, and other relevant information, were deposited in the database”. Where? how I access these information? Please, could you provide a screenshot as supplementary material?

Reply 16: We have added the URL of microRNA search to the article (line 151), and also added the detail screenshot to the help page (http://www.grapeworld.cn/gt/help.html).

Question 17: Please, revise carefully the references. Reference 25, 28 refer to supplementary materials of Reference 24 and 27.

Reply 17: We have revised the reference and check all references carefully. 
